# Osteopontin in Cardiovascular Diseases

**DOI:** 10.3390/biom11071047

**Published:** 2021-07-16

**Authors:** Kohsuke Shirakawa, Motoaki Sano

**Affiliations:** 1Department of Cardiovascular Medicine, Graduate School of Medicine, Juntendo University, Bunkyo-ku, Tokyo 1138421, Japan; shirakawa19840905@gmail.com; 2Department of Cardiology, Keio University School of Medicine, Shinjuku-ku, Tokyo 1608582, Japan

**Keywords:** osteopontin, inflammation, cardiovascular disease

## Abstract

Unprecedented advances in secondary prevention have greatly improved the prognosis of cardiovascular diseases (CVDs); however, CVDs remain a leading cause of death globally. These findings suggest the need to reconsider cardiovascular risk and optimal medical therapy. Numerous studies have shown that inflammation, pro-thrombotic factors, and gene mutations are focused not only on cardiovascular residual risk but also as the next therapeutic target for CVDs. Furthermore, recent clinical trials, such as the Canakinumab Anti-inflammatory Thrombosis Outcomes Study trial, showed the possibility of anti-inflammatory therapy for patients with CVDs. Osteopontin (OPN) is a matricellular protein that mediates diverse biological functions and is involved in a number of pathological states in CVDs. OPN has a two-faced phenotype that is dependent on the pathological state. Acute increases in OPN have protective roles, including wound healing, neovascularization, and amelioration of vascular calcification. By contrast, chronic increases in OPN predict poor prognosis of a major adverse cardiovascular event independent of conventional cardiovascular risk factors. Thus, OPN can be a therapeutic target for CVDs but is not clinically available. In this review, we discuss the role of OPN in the development of CVDs and its potential as a therapeutic target.

## 1. Introduction

Recent advances in preventive medicine have greatly improved the prognosis of cardiovascular diseases (CVDs) [1,2]. Current standard therapies for secondary prevention include the use of antiplatelet agents, lifestyle modifications, optimal medical therapy, and coronary revascularization. Despite unprecedented advances in secondary prevention, CVDs remain the leading cause of death worldwide. This suggests that the risks of CVDs are underestimated and current optimal medical therapy for secondary prevention is required for patients with CVDs.

Recently, beyond classical cardiovascular risk factors, other drivers, including inflammation, pro-thrombotic factors, and gene mutations, have been focused on not only the residual risk but also the next therapeutic target for CVDs [1,3,4]. Several biological processes contribute to the pathogenesis of CVD. In particular, the targeting of inflammatory processes in experimental animal models has been demonstrated to be beneficial in promoting healing and attenuating myocardial injury [5,6,7]. Furthermore, recent clinical trials have revealed that inflammatory activity contributes to an increased risk of CVDs in humans [8,9]. For instance, increased levels of high-sensitivity C-reactive protein (hsCRP) have been recognized as an independent predictor of both recurrent ischemia and death among patients with coronary artery disease (CAD) [10,11,12] and the reduction in hsCRP by rosuvastatin significantly reduced the incidence of major cardiovascular events (MACE) in healthy controls without hyperlipidemia [13]. In the Canakinumab Anti-inflammatory Thrombosis Outcomes Study (CANTOS) trial, canakinumab, which is an IL-1β inhibitor, resulted in a lower incidence of recurrent non-fatal myocardial infarction (MI), non-fatal stroke, or cardiovascular death among patients with stable CAD and hsCRP levels of >2 mg/L [9]. Lowering of hsCRP levels results in a reduction in MACE, cardiovascular death, and all-cause mortality without any effect on LDL-C [9]. Furthermore, the Colchicine Cardiovascular Outcomes Trial demonstrated a reduction in the primary composite outcome of cardiovascular death, cardiac arrest, non-fatal MI, stroke, or angina resulting in revascularization in patients with CAD [8]. Low-Dose Colchicine 2 (LoDoCo2), which is a proteomic substudy, revealed that colchicine treatment attenuated the NLRP3 inflammasome pathway, including interleukin (IL)-18, IL1 receptor antagonist, and IL-6 in patients with chronic CAD [14]. These data suggest that inflammation contributes to the pathogenesis of CVDs and that anti-inflammatory therapy is clinically applicable to humans.

Osteopontin (OPN) is a matricellular protein that mediates diverse biological functions [15,16,17]. OPN functions as a proinflammatory cytokine and promotes cell-mediated immune responses [18,19,20,21]. In addition to its inflammatory functions, OPN also has protective functions such as biomineralization [20,22,23] and wound healing [16,24,25,26]. OPN is involved in a number of diseases, including MI, atherosclerosis, kidney injury, diabetes, and other chronic inflammatory diseases in experimental animal models [15,19,24,25,27,28,29] and is also a strong predictor of adverse outcomes in patients with CVDs [30,31,32]. Thus, OPN is not only a risk factor but also a potential therapeutic target for CVDs. This review will explore the evidence linking OPN to CVDs and potential future therapeutic options to attenuate inflammation as a residual cardiovascular risk factor.

## 2. Diversity of the Origin and Regulation of OPN in CVDs

OPN has a two-faced phenotype dependent on the pathological state of a number of CVDs. OPN is hardly expressed under physiological conditions [24,33], but its expression markedly increases under a number of pathological conditions [24,33,34,35,36,37]. The major source of OPN varies depending on the organ and pathological conditions. Acute increases in OPN have protective roles, including wound healing and neovascularization [15,24]. In contrast, chronic increases in OPN predict poor prognosis of a major adverse cardiovascular event independent of conventional risk factors [16,30,31,36,38,39].

### 2.1. The Source of OPN

In the human heart after MI, macrophages infiltrating the infarcted myocardium make up the major source of OPN [24,40]. We previously showed that OPN is hardly expressed in normal hearts, but its expression increases markedly after MI. The hearts in Spp1 (encoding OPN) knockout (KO) mice have fewer single collagen filaments compared to the hearts in WT-sham mice, although there is no significant difference in cardiac function [33]. OPN is almost exclusively produced by galectin-3 hi CD206 + macrophages, which appear specifically in the infarcted myocardium after MI, determined by flow cytometric analysis of EGFP-Spp1-KI reporter mice heart cells subjected to MI [24]. In experimental models of chronic viral myocarditis or myocardial necrosis by transdiaphragmatic freeze-thaw, infiltrating macrophages have also been reported as the major source of OPN [34,40]. OPN has been implicated in multiple functions of macrophages, including phagocytosis, chemotaxis, angiogenesis, cytokine production, and the expression of inducible nitric oxide synthase [41]. OPN enhances phagocytosis through integrin [42] and induces macrophage migration via interaction of the C-terminal fragment with CD44 surface receptors [43] and via the SLAYGLR domain with integrin receptors [44]. OPN secreted by alternatively activated macrophages directly contributes to the phagocytic clearance of dead cells and to the reparative fibrotic response in wound healing [27,45,46,47,48,49]. OPN also induces the differentiation of cardiac fibroblasts into α-smooth muscle actin^+^ myofibroblasts [50]. Spp1 KO mice showed exaggerated post-MI left ventricular (LV) chamber dilatation associated with reduced collagen accumulation and no increase in collagen 1(α1) mRNA in infarcted myocardium after MI [24,33].

Cardiomyocytes have also been reported as a major source of OPN in diseased hearts. OPN expression is elevated in hypertrophied myocardium [51], where it modulates the activation of p38 kinase and JNKs and the development of myocardial hypertrophy in response to chronic pressure overload in mice [52]. The increase in OPN expression during streptozotocin-induced diabetic cardiomyopathy has been reported to induce cardiomyocyte apoptosis, hypertrophy, and fibrosis by modulating protein kinase C activity. Genetic depletion of OPN improves cardiac hypertrophy, function, and fibrosis in these mice [53].

Increased OPN expression protects the post-MI heart and plays a beneficial role in regulating LV remodeling after MI by promoting collagen synthesis and accumulation [24,33]. The extracellular matrix (ECM) plays an important role in the pathological remodeling of the myocardium [54,55,56]. ECM remodeling is complicated since it contributes to the development of LV dysfunction and the progression of heart failure [54,55,56]. Cardiac fibroblasts play a fundamental role in tissue remodeling after MI by modulating ECM deposition [57,58,59]. OPN in cardiac fibroblasts plays an essential role in tissue repair by enhancing their proliferative and adhesive functions. Cardiac fibroblasts derived from OPN KO mice exhibit a reduction in adhesion to ECM proteins such as collagen I compared with those derived from wild-type (WT) mice [60,61]. OPN induces cardiac fibroblasts to differentiate into α-smooth muscle actin^+^ myofibroblasts [50]. In addition, the arginine-glycine-aspartate (RGD) sequence of OPN binds to beta3 integrins on the fibroblast to promote fibroblast binding to collagen [61]. Furthermore, OPN expression is increased not only in macrophages but also in smooth muscle and endothelial cells in human coronary atherosclerotic plaques [62].

Thus, OPN is involved in a number of CVDs and the major source and role of OPN depends on pathological states. In order to use OPN as a therapeutic target, it is necessary to understand the induction and mechanism of action of OPN in each pathological state.

### 2.2. Inducer of OPN

A number of stimuli have been shown to promote OPN expression, including cytokines, reactive oxygen species (ROS), angiotensin II (Ang II), high glucose, and hypoxia [24,28,63,64,65].

We previously reported that the IL-10-STAT3-galectin-3 axis is essential for Spp1 transcriptional activation in cardiac macrophages after MI [24]. IL-10 and M-CSF act synergistically to activate STAT3 and ERK in cardiac macrophages, which in turn upregulate the expression of galectin-3 and MerTK, resulting in the functional maturation of OPN-producing macrophages [65].

Ang II upregulated Spp1 expression in adult rat cardiac fibroblasts by ROS-mediated activation of ERK1/2 and JNK pathways and IL-1beta and TNF-alpha further act synergistically with Ang II to increase Spp1 expression [64]. Ang II activates the mitogen-activated protein kinase (p42/44 MAPK) pathway followed by the induction of Spp1 expression in cardiac microvascular endothelial cells [64]. Interestingly, blocking Ang-II inhibits Spp1 expression in non-infarcted myocardium of post-MI hearts [63] and olmesartan, which is an Ang II receptor blocker, significantly decreases plasma OPN levels in patients with hypertension [66].

Furthermore, hyperglycemia and hypoxia synergistically increase OPN expression in smooth muscle cells or proximal tubule epithelial cells in the kidney [28,67]. Oxidized low-density lipoprotein (LDL) promotes proliferation and migration of smooth muscle cells derived from the human coronary artery via upregulation of OPN and MMP-9 [68]. Glucocorticoid dexamethasone has also been shown to increase Spp1 expression in cardiomyocytes and microvascular endothelial cells in vitro and OPN suppressed NOS2 activity by regulating the location and extent of NOS2 induction in these cells [69].

Thus, mechanisms that regulate the expression of Spp1 varies among cell types and pathologies, which results in the complexity of OPN as a therapeutic target for CVD.

### 2.3. OPN Isoforms

Multiple OPN isoforms exist in human as the result of alternative splicing and these isoforms have distinct biologic functions [70]. Full-length OPN can be modified by thrombin cleavage. Thrombin-cleaved OPN exposes an epitope for integrin receptors of α4β1, α9β1, and α9β4. OPN isoform includes OPNa (the full-length isoform), OPNb (which lacks exon 5 and 3) and OPNc (which lacks exon 4) [15]. In CVDs, the question of which OPN isoforms are involved or what roles each isoform have with respect to their involvement in pathological states remain elusive and thus further studies are required.

## 3. OPN in CVDs

Accumulating evidence suggests that OPN is a potent biomarker and mediator in CVDs. In this section, we discuss the roles of OPN in CVDs, specifically the ischemic heart diseases, hypertension, heart failure, dilated cardiomyopathy, atherosclerosis, and several cardiomyopathies (Appendix A).

### 3.1. Ischemic Heart Diseases

Plasma OPN levels drastically changed in a time-dependent manner in patients who underwent successful reperfusion after anterior-wall acute MI and they began to increase on day two, peaking on day three, and persisting until day 14 (420 ng/mL for control subjects versus 1139 ng/mL for day three) [71]. These data are consistent with temporal Spp1 expression in an experimental animal model [24]. We analyzed EGFP-Spp1-KI reporter mice hearts subjected to MI and GFP (the activity of Spp1 transcription in situ) was expressed almost exclusively in cardiac macrophages of the infarcted myocardium. Temporal Spp1 expression in cardiac macrophages peaked on day three and OPN secreted by cardiac macrophages directly contributed to the phagocytic clearance of dead cells and the reparative fibrotic response in wound healing after MI [24].

Plasma OPN levels also increased in patients with stable CAD [72,73] and LV ejection fraction (EF) was inversely correlated with plasma OPN levels in these patients [71,72,74]. Furthermore, in patients with stable CAD with preserved EF on optimal medical therapy, plasma OPN levels were independently associated with adverse cardiovascular outcomes [30]. In patients with a previous anterior wall MI, OPN is assumed to be released from the heart into the coronary circulation [72]. However, in a mouse model of acute MI caused by coronary artery ligation, we confirmed that there is little transcriptional activation of Spp1 in the heart 28 days after MI (Shirakawa et al., unpublished data). In patients with post-MI, OPN may continue to be released continuously from the residual coronary plaque.

Spp1 KO mice showed exaggerated post-MI LV chamber dilatation associated with reduced collagen accumulation in the infarcted myocardium after MI [24,27,33]. Cardiac fibroblasts play a fundamental role in the tissue remodeling process after MI by modulating the extent and composition of the ECM. Transforming growth factor-β1 (TGF-β1) induces the differentiation of fibroblasts to myofibroblasts in WT mice, but this does not occur in Spp1 KO mice. Cardiac fibroblasts derived from Spp1 KO mice showed a reduction in stress fibers, focal adhesions, spreading ability, and resistance to detachment by shear stress compared to those derived from WT mice [50]. These results suggest that OPN is required for myofibroblast differentiation induced by TGF-β1.

Matrix metalloproteinases (MMPs) are members of the ECM proteases and play a fundamental role in tissue remodeling in a number of CVDs, including pressure overload-induced hypertrophy, MI, or dilated cardiomyopathy (DCM) [50,75,76]. MMPs are responsible for collagen degradation in the ECM and the activation of MMP induces a reduction in cardiac tissue tensile strength and causes systolic and diastolic dysfunction [77]. OPN causes the upregulation of tissue inhibitors of MMP and collagen and the downregulation of MMP-1 expression in cardiac fibroblasts [78]. OPN inhibits IL-1β-induced activation of MMP-2 and MMP-9 via the involvement of β3 integrins and activation of protein kinase C-ζ in adult rat cardiac fibroblasts and this resulted in enhanced collagen deposition after MI [79]. MMP inhibition decreases post-MI LV dilation in Spp1 KO mice, but not in WT mice [45]. Furthermore, cardiac transgenic MMP-2 expression induces impaired contractility [80] and the deletion of MMP-2 attenuates cardiac rupture and LV dysfunction after MI [81].

OPN is also reported to be involved in angiogenesis, which is a critical and beneficial process in post-ischemic repair, after MI. Angiogenesis enhances the oxygen and nutrient supply necessary for the wound healing process and decreases the severity of ischemic events [82,83]. OPN plays an important role in post-ischemic neovascularization [84,85] and is upregulated in patients with peripheral artery disease or diabetes mellitus [86,87]. Genetic depletion of Spp1 reduces angiogenesis in cardiovascular endothelial cells after MI. OPN KO mice show impaired myocardial angiogenic response and results in adverse remodeling post-MI [88].

Thus, OPN has a complex effect on cardiac remodeling after MI (Figure 1). Inhibition of OPN may not only ameliorate excessive fibrosis, but also promote MMP activity and inhibit angiogenesis. Furthermore, the major source of OPN in the chronic phase of MI remains unclear. There are still many issues that need to be resolved to achieve secondary prevention of ischemic heart disease by targeting OPN.

### 3.2. Hypertension

An increase in OPN expression in aortic tissues has been observed in a number of animal models of hypertension and its expression is correlated with systolic blood pressure [89]. OPN expression is increased via the Akt1/AP-1 dependent pathway and promotes the production of MMP-2 in vascular smooth muscle cells followed by vascular remodeling [89]. The effects of OPN on MMP are opposite in cardiac fibroblasts and vascular smooth muscle cells and the action of OPN may be target cell-dependent. Furthermore, OPN plays an essential role in modulating compensatory cardiac hypertrophy in response to chronic pressure overload and MMP regulation by OPN is important for this process [52].

### 3.3. Heart Failure

Plasma OPN levels were elevated in patients with chronic heart failure (HF) due to LV dysfunction and correlated with the severity as assessed by the New York Heart Association class (NYHA). Furthermore, plasma OPN levels predicted death within four years of follow-up in these patients (382 ng/mL for control subjects versus 532 ng/mL for patients with HF and 479 ng/mL for the New York Heart Association (NYHA) class I/II versus 672 ng/mL for NYHA class III/IV) [90]. OPN expression is increased in the myocardium of patients with hypertensive heart disease and HF compared to the controls and the increase in OPN expression is correlated with lysyl oxidase, insoluble collagen, pulmonary capillary wedge pressure, and LV chamber stiffness while being inversely correlated with LVEF [91]. Furthermore, serum levels of OPN are predictors of ventricular tachycardia and fibrillation in patients with HF [92].

Col4a3^−/−^ mice show a phenotype similar to that of heart failure with preserved EF, including diastolic dysfunction, cardiac hypertrophy and fibrosis, pulmonary edema, and impaired mitochondrial function. Genetic depletion of OPN ameliorates these phenotypes by regulating mitochondrial oxoglutarate dehydrogenase L [32].

However, the major source of OPN remains unclear. HF is a complex multiple organ disease and thus OPN production may increase not only in the heart but also in other organs, including the kidney, lung, and liver. Elucidation of the major source and inducer of OPN results in novel therapies for HF.

### 3.4. Dilated Cardiomyopathy

Plasma OPN levels are elevated and associated with the severity of HF, as assessed by NYHA class in patients with DCM [32]. Analysis of the myocardium obtained from patients with DCM revealed that the main source of OPN was cardiomyocytes and the extent of Spp1 expression showed a positive correlation with LV end-systolic diameter and a negative correlation with LVEF in these patients [93]. Furthermore, Spp1 expression levels in the myocardium are elevated, especially in patients with a LVEF of <50% and correlated with the accumulation of collagen type I [94], serving as a biomarker for prognosis in patients with DCM [95].

Cardiac-specific integrin-linked kinase (ILK) is a serine/threonine kinase that has been linked to HF. ILK KO mice spontaneously develop phenotypes such as DCM, including apoptosis, fibrosis, and inflammation. Blocking OPN attenuates the decline in cardiac function in these mice [96]. Furthermore, the inhibition of OPN suppressed collagen I secretion in fibroblasts through a FAK/Akt-dependent pathway and reduced LV remodeling and dysfunction in mice with DCM [97]. Thus, although effective treatments for DCM are scarce, OPN may be a future therapeutic target for DCM.

### 3.5. Atherosclerosis

OPN is primarily expressed in endothelial cells, macrophages, and smooth muscle-derived foam cells, with an increased expression in human atherosclerotic plaques in the aorta, carotid, and coronary arteries [62,98]. The increase in OPN expression in plaques is reported to be correlated with the formation of ulceration, inflammation, and unstable plaques in patients who underwent carotid endarterectomy [99]. Serum OPN levels were higher in patients with acute coronary syndrome than in those with stable CAD and were associated with rapid coronary plaque progression and in-stent restenosis (684 ng/mL for no progression versus 977 ng/mL for progression) [100]. On the contrary, it has been reported that OPN expression levels of coronary plaque in patients who underwent endarterectomy during coronary bypass surgery are lower in unstable plaques and plaques with large calcification deposits [101].

In mice, the overexpression of OPN induced the development of medial thickening and neointimal formation and OPN KO mice showed attenuation of atherosclerosis [102]. The overexpression of OPN develops early fatty-streak and mononuclear cell-rich lesions in a high-fat diet (HFD)-induced atherosclerosis model of OPN transgenic mice [103]. Thus, OPN plays a fundamental role in the development and progression of atherosclerotic plaques.

### 3.6. Cardiovascular Calcification

OPN is recognized as an inhibitor of mineral deposition in the vascular wall and cardiac valves [15]. OPN was significantly and positively correlated to the coronary calcium score evaluated by cardiac computed tomography [104]. Smooth muscle cells isolated from the aortas of Spp1 KO mice showed higher calcification compared to those of WT mice in the presence of inorganic phosphate, which is an inducer of vascular calcification [105]. Furthermore, OPN phosphorylation is required for the inhibition of human vascular smooth muscle cell calcification [106,107]. Epigenetic modifications of the genome, such as DNA methylation and histone modifications, play an important role in the development of CVDs [108,109]. For instance, H3K9a, H3K27ac, or H3K4 have been reported to increase in atherosclerotic smooth muscle cells or macrophages [110]. There are several reports on epigenetic regulation of OPN. OPN is markedly upregulated in the alveolar macrophages of the bleomycin model of lung fibrosis. The HIF1α-driven glycolysis in alveolar macrophages altered the epigenetic modification of OPN, with the metabolic intermediate 3-phosphoglyceric acid, and reduced its expression [111]. Epigenetic regulations, including Methyl-CpG binding protein 2 and the activation of nuclear calcium signaling, have been demonstrated to be involved in OPNc production [112]. Epigenetic regulation of Spp1 in CVDs remain unclear and thus further studies are required. Hydroxyapatite deposition in atherosclerotic coronary plaques is associated with OPN expression [113]. OPN has been reported to inhibit calcification by blocking hydroxyapatite crystal formation and regulating acidification, while carbonic anhydrase II in macrophages and implanted glutaraldehyde-fixed aortic valve leaflets showed accelerated calcification in OPN KO mice compared to WT mice [114,115]. Exogenous recombinant histidine-fused OPN, which is phosphorylated and contains a functional RGD cell adhesive domain, attenuated ectopic calcification of glutaraldehyde-fixed bovine pericardium in rats [115]. Furthermore, OPN promotes the differentiation of monocytes derived from patients with vascular calcification into alternatively activated macrophages and inhibits their differentiation into osteoclasts [116].

In addition, foamy macrophages persist in atherosclerotic plaques and promote disease progression [117]. In the context of atherosclerosis, OPN is generally regarded as a proinflammatory and proatherogenic molecule. A significantly higher expression of OPN was observed in foam cells of atherosclerotic plaques [118,119]. However, recent single-cell RNA sequencing analysis revealed that non-foamy macrophages formed a number of inflammatory transcripts, including IL-1β, and foamy macrophages expressed fewer inflammatory genes than non-foamy macrophages in the atherosclerotic aorta [120,121]. These data suggest that OPN-producing macrophages may have a protective effect on vascular calcification in atherosclerotic plaques. OPN promotes atherosclerosis but inhibits the calcification. This is expected to be dependent on the type of OPN and the localization of OPN-producing cells in atherosclerotic plaques.

### 3.7. Other Heart Diseases

OPN is involved in the pathogenesis of CVDs. Cardiac-specific overexpression of OPN showed cardiomyocyte loss, severe fibrosis, and Th1 cell infiltration followed by LV diastolic and systolic function and the development of myocarditis in mice [122]. Furthermore, OPN expression is increased in macrophages, resulting in cardiac fibrosis and enhanced cardiac remodeling in a model of coxsackievirus B3-induced myocarditis [123]. MMP-9 is increased in the hearts of dystrophin-deficient mdx mice, which possess point mutation in the Duchenne muscular dystrophy-related gene, and genetic ablation of MMP9 diminishes the activation of ERK1/2 and Akt kinase followed by an improvement in cardiac function [124]. OPN has been shown to enhance MMP-9 expression in cardiac and skeletal muscle, resulting in the development of myopathy in these mice. Furthermore, OPN induces myocardial MMP-2 activation followed by the development of cardiac remodeling, inflammation, and interstitial fibrosis in a mouse model of chronic Chagas heart disease [125]. In patients with amyloid light-chain amyloidosis, serum OPN levels are an independent predictor of all-cause mortality [126]. OPN/p38 MAPK signaling pathways are suggested to protect against ascending aortic aneurysm progression [127]. Thus, OPN plays an essential role in the pathogenesis of several heart diseases.

## 4. OPN in Multi-Organ Crosstalk

Multi-organ crosstalk is defined as a complex biological interaction between distant organs. CVDs are not only single organ diseases, but rather a complex multi-organ syndrome with impairments including the kidney, endocrine, lung, skeletal muscle, liver, and hematopoietic system. Many molecules, including proinflammatory cytokines and growth factors, are reported to be involved in multi-organ dysfunction in CVDs. In particular, the interplay between the heart, kidney, and diabetes plays an essential role in the pathogenesis of CVD. In this section, we review the role of OPN in the kidney, diabetes, and cardiorenal syndrome.

### 4.1. OPN in the Kidney

In both humans and mice, OPN is reported to be mainly expressed in the thick ascending limb of the loops of Henle and distal nephrons in the steady state [19]. The physiological role of OPN in normal kidneys remains unclear and the transcriptional activation of OPN rarely occurs in normal kidneys [19]. Furthermore, Spp1 KO mice showed no abnormalities in the kidney. In contrast, in a number of pathological conditions, OPN is highly upregulated and plays a critical role in interstitial fibrosis, especially in the proximal tubules [128]. Multiple factors, including cytokines, hormones, hyperglycemia, and hypoxia, induce OPN upregulation in the kidneys [128]. Tubular injury can result in abnormalities in many metabolic and signaling pathways, which contribute to tubulointerstitial fibrosis and kidney dysfunction. The different cell types and pathological states differ in their regulatory mechanisms of Spp1 expression in the kidney.

#### 4.1.1. Acute Kidney Injury

The plasma OPN levels were elevated and independently predicted the 28 day survival rate in patients with acute kidney injury (AKI) [129]. Urine levels of OPN at admission and on day three were associated with an increased risk of AKI and mortality in survivors after out-of-hospital cardiac arrest [130]. OPN is also recognized as a predictor of early kidney transplant rejection [131]. OPN can activate NK cells to mediate the apoptosis of the tubular epithelial cells [132]. Blockade of OPN inhibited the infiltration of NK cells into the kidney, resulting in a reduction in the expression of injury markers, including neutrophil gelatinase-associated lipocalin and kidney injury molecule-1, and apoptosis of renal tissue in mice after renal ischemia reperfusion injury [133]. Genetic depletion of Spp1 results in a reduction in apoptosis in both the tubular epithelium and interstitium in the acute phase of ischemia-reperfusion injury in mice [134]. Furthermore, following ischemia-reperfusion injury, OPN KO mice showed less macrophage infiltration and interstitial fibrosis compared to WT mice [134]. These results suggest that inflammation and oxidative stress induced by ischemia-reperfusion injury may result in the upregulation of OPN, which causes renal damage in AKI.

#### 4.1.2. Chronic Kidney Injury

OPN is involved in AKI as well as chronic kidney disease (CKD). The plasma OPN levels, but not urine levels, were significantly higher in diabetic patients than in healthy controls [95,135]. The increase in plasma levels of OPN is associated with the development of microalbuminuria and negatively correlated with eGFR in patients with diabetes [136,137]. In experimental diabetic animal models, OPN expression in the proximal tubules was also reported to be markedly upregulated in db/db and streptozocin-induced diabetic mice [138,139]. Hyperglycemic conditions induce histone acetylation and methylation, which in turn results in upregulation of OPN gene expression [138]. Furthermore, Ang II accelerates macrophage infiltration and fibrosis in the kidney and subsequently results in the elevation of blood pressure and the increase in urinary albumin/creatinine ratios in mice. Genetic depletion of OPN protects Ang II-induced inflammation, oxidative stress, and fibrosis of the kidney [140,141]. Thus, OPN has been recognized as a facilitator of tubulointerstitial injury in the pathogenesis of CKD.

### 4.2. Diabetes

As previously mentioned, OPN plays an essential role in the pathogenesis of diabetic neuropathy in mice [138,139]. OPN has been reported to be increased in the circulating blood of obese diabetic and insulin-resistant patients [136,142] and to enhance visceral adipose inflammation, which results in insulin resistance [143,144]. We previously reported that a CD153^+^PD-1^+^CD44^hi^CD4^+^T cell population, which accumulates in the visceral adipose tissue of HFD-induced obese mice, causes adipose inflammation by producing large amounts of OPN [29]. Interestingly, CD153^+^PD-1^+^CD44^hi^CD4^+^T cells acquire a senescent phenotype and are not easily eliminated as a negative legacy of obesity after weight reduction [145]. Furthermore, it has been reported that OPN expression increases in the epicardial fat from patients with CAD compared to controls and is associated with the presence of calcified atherosclerotic plaques [146]. Flow cytometric analysis of epicardial fat in HFD-induced obese EGFP-Spp1-KI reporter mice demonstrated that CD153^+^PD-1^+^CD44^hi^CD4^+^T cells markedly increased and produced a large amount of OPN (Shirakawa K., unpublished observation). These findings suggest that senescent-adipose CD4 T cells in epicardial fat may play a pivotal role in the development of CVDs in patients with diabetes.

### 4.3. Cardiorenal Syndrome

Cardiorenal syndrome (CRS) is involved in a spectrum of disorders involving both the heart and kidneys and is currently the focus of much attention as a therapeutic target for HF. Acute or chronic dysfunction of the heart or kidneys can induce acute or chronic dysfunction in other organs and several factors are involved in the interactions between the heart and kidney [147]. OPN is recognized as one of the key molecules that mediates cardiorenal syndrome (Figure 2). In fact, plasma OPN and creatinine levels are elevated in patients with stable CAD. Furthermore, higher levels of OPN are associated with multivessel lesions and decline in the renal function of these patients [148]. Serum levels of OPN predict adverse cardiovascular events in patients with CKD [149]. However, the major source and role of OPN in CRS still remain elusive.

The cardioprotective effect of SGLT2 inhibition, which is responsible for the reabsorption of sodium and glucose in the proximal tubules of the kidney, has reaffirmed the significance of CRS in the pathogenesis of heart failure and CKD. Empagliflozin reduced the rates of death from cardiovascular causes, hospitalization for heart failure, and death from any cause in patients with type 2 diabetes who are at high risk for cardiovascular events and who received empagliflozin [150]. Canagliflozin also reduced the risk of cardiovascular death and hospitalization for heart failure in patients with type 2 diabetes with a high risk of CVD [151]. Furthermore, dapagliflozin resulted in a lower rate of cardiovascular death or hospitalization for HF in patients with type 2 diabetes and who had no atherosclerotic CVDs [152]. More recently, among patients with HF and a reduced EF, the risk of worsening HF or death was lower in the SGLT2 inhibitor group than in the placebo group, regardless of the presence or absence of diabetes [153]. These results suggest a beneficial effect of SGLT2 inhibition beyond glycemic control. However, the reason for the prognostic improvement of CVD with kidney-targeted treatment remains unclear.

Under healthy conditions, aerobic energy metabolism in the proximal tubule occurs with fatty acids as the energy substrate [154]. A recent systematic approach with transcriptomics, metabolomics, and metabolic flux analysis demonstrated that metabolic reprograming has occurred in proximal tubular epithelial cells in diabetes [155,156,157]; specifically, oxygen consumption is increased, glycolytic pathways are set in motion, and the production of ROS in mitochondria is enhanced [158]. Increased oxygen consumption in tubular epithelial cells results in a decrease in the partial pressure of tissue oxygen in the renal cortex (hypoxia) [159]. These changes were responsible for tubulointerstitial fibrosis. The SGLT2 inhibitors decrease oxygen consumption in proximal tubular epithelial cells and improve tissue oxygenation [159]. We found that in cultured proximal tubular epithelial cells, a high-glucose environment drives the glycolytic pathway and induces enhanced mitochondrial ROS and transcriptional activity of OPN [28], which could be suppressed by inhibiting the glycolytic pathway or by inhibiting SGLT2-mediated uptake of glucose into the cell [28]. In diabetes mellitus, myoinositol oxygenase (MIOX) is overexpressed in proximal tubular epithelial cells and activates the myoinositol pathway, which is a collateral pathway of glycolytic pathways [160]. This results in mitochondrial fragmentation and depolarization and simultaneously impairs mitophagy [160]. The resulting accumulation of impaired mitochondria induces excessive ROS production and cell death [161]. We found that SGLT2 inhibitors suppress the increased expression of MIOX in a high-glucose environment [28]. These results suggest that SGLT2 inhibitors may prevent the development of heart failure via CRS as a result of ameliorating abnormal glucose metabolism in the proximal tubules of the kidney and suppresses OPN production [162].

## 5. Conclusions

Accumulating evidence demonstrates the significance of OPN in the pathogenesis of CVD. The effects of OPN are disease-specific and consists of positive and negative effects. Therefore, OPN is a useful biomarker for predicting the risk of CVD; however, OPN itself cannot be directly targeted for treatment. Instead, the mechanism of OPN production specific to each condition should be the target of therapy.

## Figures and Tables

**Figure 1 biomolecules-11-01047-f001:**
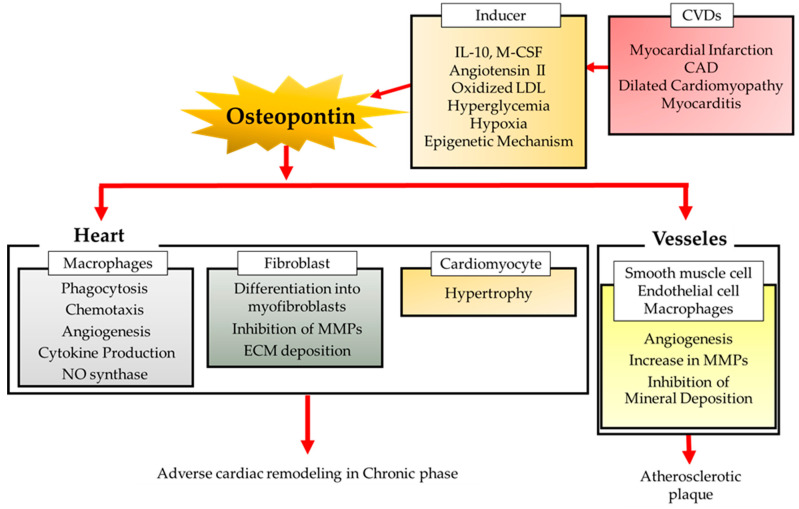
OPN in cardiac remodeling. OPN is involved in a number of CVDs, including ischemic heart diseases, hypertension, heart failure, dilated cardiomyopathy, atherosclerosis, and several cardiomyopathies and the major source and role of OPN is dependent on pathological states. A number of stimuli are shown to promote OPN expression, including cytokines, ROS, Ang II (angiotensin II), high glucose, hypoxia, and epigenetic mechanisms. Different cell types and pathological states differ in their regulatory mechanisms of OPN in the heart and vessels.

**Figure 2 biomolecules-11-01047-f002:**
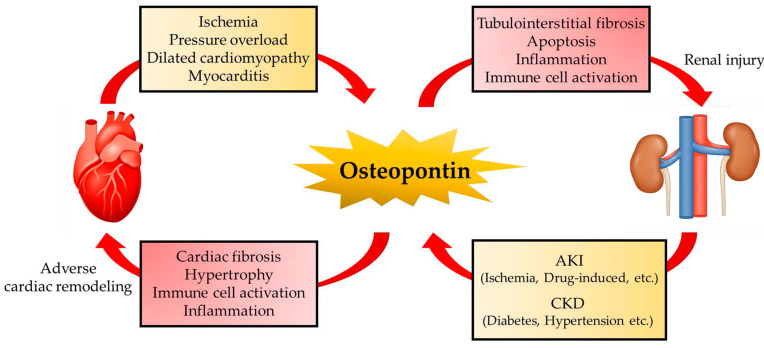
Vicious cycle of OPN in CRS. OPN is involved in the pathogenesis of CRS. Acute or chronic dysfunction of the heart or kidneys can induce OPN expression and may result in acute or chronic dysfunction in the other organ. Several stimuli induce OPN in the pathological conditions of the heart and kidney and OPN causes cardiac fibrosis, cardiac hypertrophy, LV dysfunction, and tubulointerstitial fibrosis.

## Data Availability

Not applicable.

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
