# Peer review of "Osteopontin in Cardiovascular Diseases"

_biomolecules, 2021, doi:10.3390/biom11071047_

Round 1

Reviewer 1 Report

Thank you for the opportunity to read this well-presented narrative review. It was a pleasure to read and the flow and layout of the section were easy to follow and logical. The citations are sufficiently current. A few minor points for improving the utility of the work: -levels of osteopontin- when discussing plasma levels 'baseline/normal' or 'increases' (experimental animals/species, human-specific disease condition), it would be helpful if some reference range indications from the cited literature would be helpful, ideally even a table citing their source publications would be useful for the reader to refer to. -osteopontin pharmacodynamics- are there details to provide regarding, the timing of cellular increase post-injury/stress, clearance rate, and proteolysis/cleavage in plasma? - osteopontin protein - are there different isoforms? Role in development/cell differentiation? - Figure - useful schemes but please delete 'etc'

Author Response

Thank you for your constructive suggestion.

We have added the plasma concentrations of OPN in each pathological condition. Plasma cleavage mechanisms and isoforms of OPN have been also added, as well as the role of OPN in cardiac development in revised manuscript.

Added sentence in revised manuscript

Multiple OPN isoforms exist in human as the result of alternative splicing and these isoforms have distinct biologic functions. Full-length OPN can be modified by thrombin cleavage. Thrombin-cleaved OPN exposes an epitope for integrin receptors of α4β1, α9β1, and α9β4. OPN isoform includes OPNa (the full-length isoform), OPNb (which lacks exon 5, and 3) and OPNc (which lacks exon 4). In CVDs, what OPN isoforms or what role of each isoform are involved in each pathological states remained elusive, so further studies are required.

Heart in Spp1-KO sham mice have fewer single collagen filaments compared with those in WT-sham mice, although there is no significant difference in cardiac function.

Reviewer 2 Report

I would better update the bibliography even with more recent works we are in 2021 and there are several works published on osteopontin, furthermore I would add a third figure or table where to summarize the involvement of osteopontin in the different types of pathways in cardiovascular pathology

Author Response

Thank you for your constructive suggestion.

We have added some literature on osteopontin published in 2021. Furthermore, we have also added a table of OPN-related CVDs.

Added sentence in revised manuscript

OPN/p38 MAPK signaling pathways is suggested to protect against ascending aortic aneurysm progression.

OPN was significantly and positively correlated to coronary calcium score evaluated by cardiac computed tomography.

Reviewer 3 Report

The authors presented an interesting review about the role of osteopontin (OPN) in cardiovascular diseases. Many aspects related to OPN sources and pathophysiological effects were well described. The correlation of OPN, cardiovascular diseases, and multiple organ cross-talk was elegantly reported and it was able to arouse in the reader for a different point of view in the treatment of cardiovascular dysfunctions.

The authors could explore with more details some aspects of intracellular signaling related to OPN activation in hypertension and in heart failure. It seemed that OPN and matrix metalloproteinases (MMPs) have different regulations depending on cell type. The authors could show more published results related to this cross-talk with OPN and MMPs in cardiovascular and renal cells. Regarding OPN post-translational modification, the authors commented in line 281 that OPN could be phosphorylated in the inhibitor process of human vascular smooth muscle cell calcification. However, this reviewer would like to know more information about this post-translational modification in the OPN molecule. Is this process altered in cardiovascular diseases? Can OPN suffer another type of post-translational modification and what could be these effects?

Author Response

Thank you for your constructive suggestion.

Epigenetic modifications of the genome, such as DNA methylation and histone modifications, play an important role in the development of CVDs. Epigenetic regulation of Spp1 in CVDs remain elusive, so further studies are required. Thus, we discussed about epigenetic regulation of Spp1 in other disease model and possible involvement in CVDs in revised manuscript as below.

Added sentence in revised manuscript

Epigenetic modifications of the genome, such as DNA methylation and histone modifications, play an important role in the development of CVDs. For instance, H3K9a, H3K27ac or H3K4 have been reported to increase in atherosclerotic smooth muscle cells or macrophages. There are several reports on epigenetic regulation of Spp1.Osteopontin is markedly upregulated in alveolar macrophages of the bleomycin model of lung fibrosis. The HIF1α-driven glycolysis in alveolar macrophages altered the epigenetic modification of Spp1, with the metabolic intermediate 3-phosphoglyceric acid, and reduced its expression. Epigenetic regulations, including Methyl-CpG binding protein 2 and the activation of nuclear calcium signaling, have been shown to be involved in OPNc production. Epigenetic regulation of Spp1 in CVDs remain unclear, so further studies are required.